# POLYHEDRONNET: REPRESENTATION LEARNING FOR POLYHEDRA WITH SURFACE-ATTRIBUTED GRAPH

**Dazhou Yu   Genpei Zhang   Liang Zhao** *

Department of Computer Science, Emory University

dyu62@emory.edu, genpeizhang2024@gmail.com, liang.zhao@emory.edu

## ABSTRACT

Ubiquitous geometric objects can be precisely and efficiently represented as polyhedra. The transformation of a polyhedron into a vector, known as polyhedra representation learning, is crucial for manipulating these shapes with mathematical and statistical tools for tasks like classification, clustering, and generation. Recent years have witnessed significant strides in this domain, yet most efforts focus on the vertex sequence of a polyhedron, neglecting the complex surface modeling crucial in real-world polyhedral objects. This study proposes **PolyhedronNet**, a general framework tailored for learning representations of 3D polyhedral objects. We propose the concept of the surface-attributed graph to seamlessly model the vertices, edges, faces, and their geometric interrelationships within a polyhedron. To effectively learn the representation of the entire surface-attributed graph, we first propose to break it down into local rigid representations to effectively learn each local region's relative positions against the remaining regions without geometric information loss. Subsequently, we propose PolyhedronGNN to hierarchically aggregate the local rigid representation via intra-face and inter-face geometric message passing modules, to obtain a global representation that minimizes information loss while maintaining rotation and translation invariance. Our experimental evaluations on four distinct datasets, encompassing both classification and retrieval tasks, substantiate PolyhedronNet's efficacy in capturing comprehensive and informative representations of 3D polyhedral objects. Code and data are available at *github.com/dyu62/3D_polyhedron*.

## 1   INTRODUCTION

In mathematics and computational geometry Yu et al. (2025), a polyhedron is defined as a three-dimensional (3D) solid formed by flat polygon faces joined at edges and vertices. Ubiquitous geometric shapes can be precisely and efficiently modeled as polyhedra, ranging from basic 3D shapes (e.g., cubic, pyramid, and truncated tetrahedron) to compositions of them (e.g., shapes of buildings, furniture, and digital objects in CAD) as exemplified Figure 1 (a). In the real world, there are many tasks surrounding polyhedra such as classification (e.g., convex or concave); clustering polyhedra into different types (e.g., Platonic solids and prisms ); as well as generation and optimization (e.g., use faceted facades to break up flat surfaces) of polyhedra for design needs. However, the raw form of polyhedra cannot be directly input into machine learning models which require structured formats such as vectors, tensors, etc. Hence, a fundamental upstream task is to map a polyhedron into a vector representation, namely polyhedra representation learning, which is the focus of this paper.

Recent studies on polyhedral geometries can be broadly classified into two categories. The first category involves feature engineering on the faces of a polyhedron to generate descriptors for each face and aggregate these features (Qi et al., 2017b; Shi & Rajkumar, 2020; Wang et al., 2019). However, this manual selection of features is limited and biased by human knowledge, which can result in the loss of geometric information at the initial stage and often lacks generalizability to other tasks. The second category models the shapes of polyhedral faces directly using sequences of coordinates, preserving the original geometric information and learning features from the data, which can be generalized across various tasks (Mai et al., 2023; van't Veer et al., 2019; Yan et al.,

---

*Corresponding author.

2021). Nevertheless, these methods are constrained by the need for a specific order of input and do not consider the relationship among faces. Directly using coordinates also fails to account for rotation and translation invariance, thus limiting the ability to consistently interpret polygonal geometries regardless of their spatial orientation or position. Moreover, such approaches neglect face properties, which contain significant semantic information, by focusing solely on the shapes of polygonal faces. Figure 1 (b) illustrates how face attributes introduce semantic information that influences the appearances and functionalities of geometric objects. Although they share the same underlying polyhedral structure, the three objects are distinctly different. The first polyhedron is the Louvre Pyramid, which is characterized by four glass faces, with a concrete ground face. The middle one is a wireframe pyramid with empty faces, emphasizing the geometric structure and suggesting its use as a craft or model. The last one is an Egypt pyramid, featuring yellow stone faces.

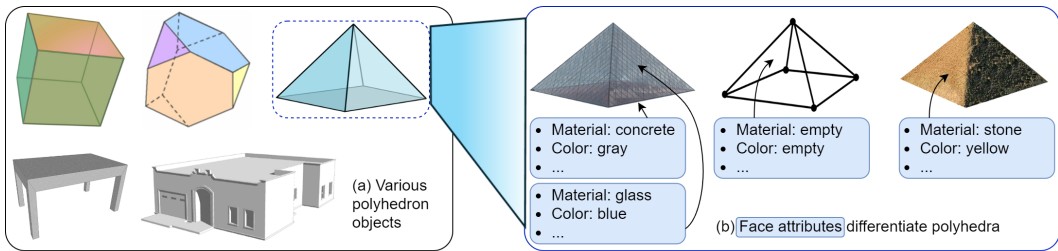

Figure 1: 3D objects modeled as polyhedra.

To address these limitations, we propose **PolyhedronNet**, a novel framework for polyhedra representation learning. Firstly, we propose the **Surface-Attributed Graph (SAG)** to concisely encapsulate the information of a polyhedron. Beyond simple graphs, SAG utilizes face-hyperedges to model the geometric relationships among vertices, edges, and faces and explicitly capture the face semantics, ensuring no information is lost. Thus, learning the representation of a polyhedron is equivalent to learning SAG representation. We solve this problem by first decomposing the SAG using the **Local Rigid Representation** of SAG and then aggregating them to SAG's global representation. In each local rigid representation, to preserve the current local region's geometric relation to the whole SAG, we calculate the second-order distances around a node and angles formed by its neighbors and associated faces to form a rigid body around the node. The set of local rigid bodies encapsulates complete geometric and semantic information in the SAG and provides rotation and translation invariance. Thirdly, we propose **PolyhedronGNN** to hierarchically aggregate the local rigid representation into a global representation that minimizes information loss while maintaining rotation and translation invariance of global representation. Considering faces are the pivots of a polyhedron, this model learns geometric information inside faces and across faces, based on the two-hop paths that suffice the preservation of local rigid information. This design adeptly captures the semantic heterogeneity of the surface-attributed graph, significantly enhancing the model's ability to uniquely identify and differentiate diverse input graphs. Moreover, we empirically validate our proposed method across four datasets and demonstrate its effectiveness in both classification and retrieval tasks, significantly outperforming state-of-the-art approaches by a substantial margin.

## 2 RELATED WORK

### 2.1 3D OBJECT REPRESENTATION LEARNING

Traditional methods render a three-dimensional object into two dimensions as an image or a set of images with different views (Qi et al., 2016; Su et al., 2015). These methods involve significant information loss and cannot truly represent 3D objects. Some recent works (Qi et al., 2017a; Le & Duan, 2018) utilize spatial point cloud to depict objects. PointNet (Qi et al., 2017a) introduced a deep learning framework for directly processing point clouds, significantly advancing object classification and segmentation tasks. This was further expanded by Le & Duan (2018) through PointGrid, which combines point clouds with voxel grids to enhance geometric understanding. Voxel grid representation offers a volumetric approach to 3D shape analysis. Chen et al. (2023) develop PolyGNN to reconstruct 3D building models using polyhedral decomposition from point cloud. Wu et al. (2015b) developed 3D ShapeNets, a method that leverages convolutional neural networks on

voxel grids to perform 3D shape recognition, providing a robust framework for capturing complex shapes. Wang et al. (2017) introduced the Octree-based CNN, which improves efficiency by using octree structures for adaptive resolution in 3D space. These discrete methods fail to leverage the structural information like edges inherently by points or grids, making them less compatible with structured data. Mesh representation focuses on using triangles or quads to model 3D objects. Bruna et al. (2013) proposed spectral networks to operate on meshes. Henaff et al. (2015) extended this concept by introducing convolutional networks for structured data, enhancing the analysis of mesh topology. Further advancements by Defferrard et al. (2016) and Monti et al. (2017) applied localized filtering and mixture model CNNs to learn geometric features on meshes. Pang et al. (2023) proposes a GNN-based approach to learn geodesic embeddings for polyhedral faces. While these methods have significantly advanced the processing of 3D object, they face limitations due to their computational intensity with high-resolution models and their struggles with irregular geometries, inherent to the mesh format. Directly modeling objects with polyhedra is a promising method to address these issues.

## 2.2 POLYHEDRAL REPRESENTATION LEARNING

Recent advancements in the field of polyhedral geometry representation learning have been significant. Traditional feature engineering approaches (Pham et al., 2010; Yan et al., 2019; He et al., 2018) transform polygonal shapes into predefined shape descriptors. GNNs are utilized to improve the handling of spatial relationships and structural complexities (Qi et al., 2017b; Shi & Rajkumar, 2020; Wang et al., 2019). However, these descriptors oversimplify the data, failing to capture the complete spectrum of shape information. Polygon shape encoding methods (van't Veer et al., 2019; Mai et al., 2023; Yan et al., 2021), have demonstrated their effectiveness in shape classification and retrieval tasks. While beneficial for certain types of analysis, these methods do not fully meet the needs of polyhedral representation learning that requires capturing complex topological relationships between polygonal geometries. PolygonGNN Yu et al. (2024) is the first GNN-based method that captures both multi-polygon relationships and individual polygon shape information. However, it is designed specifically for 2D shapes. In relation to polyline representation learning (Jiang et al., 2021; 2022), current works focus on processing continuous lines and curves that delineate the boundaries and configurations of shapes in spatial data. Another category of research focuses on polyhedron generation. Gillsjö et al. (2023) extracts polygons from images by using heterogeneous graphs and wireframes to learn feature space. Zorzi & Fraundorfer (2023) utilizes edge-aware GNNs to enhance polygon detection accuracy and applicability in scene parsing by considering both node and edge features.

## 3 PROBLEM FORMALIZATION

In this section we first introduce the formal definitions of polygons (Mai et al., 2023) and polyhedra (Weisstein), and then formalize the problem of polyhedra representation learning.

**Definition 3.1** (Polygon). A polygon $p_i$ is defined as an ordered sequence of vertices that form a closed shape: $p_i = (v_{i,1}, v_{i,2}, \ldots, v_{i,N_{b,i}})$, where $v_{i,j} \in \mathbb{R}^3$ denotes the 3D coordinates of the $j$-th vertex, $N_{b,i}$ denotes the number of vertices. The vertices of the polygon are coplanar, meaning they all lie within a single 2D plane that is embedded in 3D space. Additionally, the polygon is assumed to be simple, which implies that it does not have any self-intersections or holes.

**Definition 3.2** (Polyhedron). A polyhedron $q$ is a 3D solid that consists of a collection of polygonal faces $q = \{p_i\}_{i=1}^{N_f}$, where each face $p_i$ is a polygon as defined in Definition 3.1. The vertices of each face are ordered in a counterclockwise direction when viewed from outside the polyhedron, ensuring a consistent orientation across all faces. The normal vector associated with each face $p_i$ can be obtained using the right-hand rule, pointing outward from the polyhedron. In addition to the geometric properties, each face $p_i$ may have semantic face attributes, which can include material, color or other application-specific data.

This definition provides a unified data structure for both 2D polygons and 3D polyhedra. A polygon can be treated as a special case of a polyhedron with a single face. By defining the faces as oriented polygons, our representation implicitly captures the orientation and enclosure properties of the polyhedron.

**Polyhedra representation learning**. This paper aims to convert a polyhedron into a vector representation, denoted as $q \to q_v$, where $q_v \in R^d$ and $d$ represents the dimension of the vector. As depicted in Figure 1, face attributes collectively identify object patterns, which is fundamental to understanding the concept of a polyhedron. The learned representation $q_v$ should capture the geometric and semantic properties of the polyhedron, while being invariant to rotation and translation transformations. Furthermore, the representation should be discriminative, enabling accurate classification, retrieval, and other downstream tasks on 3D shapes.

## 4 METHODOLOGY

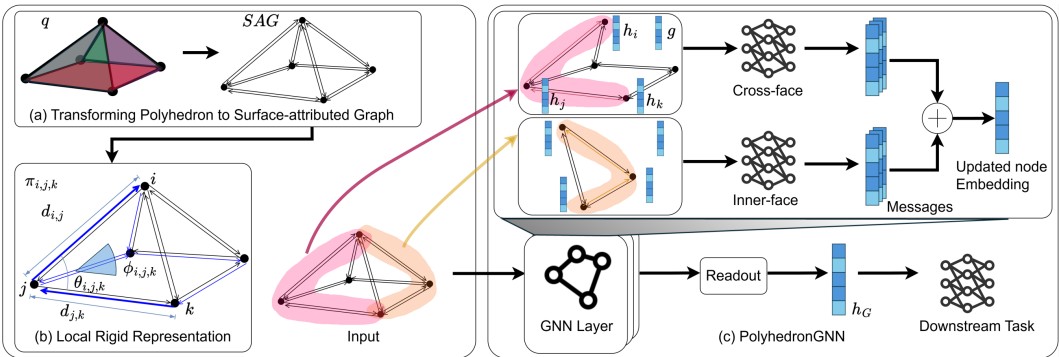

Figure 2: Illustration of the proposed framework.

To learn distinct representations for polyhedra by addressing the aforementioned challenges, we propose the PolyhedronNet framework, as shown in Figure 2. In Figure 2 (a), to unify the characterization of vertices, edges, faces, and their relationships in a polyhedron, we propose a transformation that turns a polyhedron into a surface-attributed graph (SAG), as elaborated in Section 4.1. This process is proven to be invertible, which maintains information in the polyhedron while converting it to a graph data format. In Figure 2 (b), to learn a representation of the SAG, we decompose SAG into a set of local rigids for each 2-hop path within a polyhedron (Section 4.2) with our local rigid representation. The representation is a five-tuple set that transforms absolute coordinates into vectors while preserving the original graph information and achieving rotation and translation invariance. In Figure 2 (c), we propose a novel graph neural network, PolyhedronGNN (Section 4.3), to aggregate the local rigid representations into the final SAG representation.

### 4.1 TRANSFORMING POLYHEDRON TO SURFACE-ATTRIBUTED GRAPH

Graphs provide a natural way to capture the geometric structure of a polygonal shape by representing vertices as nodes and edges as links between them. In recent years, graphs have been successfully applied for polygon-related tasks (Zhou et al., 2023; Zorzi et al., 2022; Zorzi & Fraundorfer, 2023). These studies have demonstrated the effectiveness of using graphs to capture the intricate geometric and topological properties of polygonal shapes. Given that a polyhedron can be considered as a 3D extension of a polygon, it is natural to extend the graph representation to the polyhedron domain. We let each graph node represent a vertex of a polyhedron and each directed graph edge represent an edge of a face in the polyhedron.

However, A polyhedron is characterized not only by the vertices and edges but also by the faces. Developing a comprehensive representation of polyhedra necessitates a unified data structure capable of encapsulating all the geometric information. While vertices and edges are naturally contained in a graph structure, we propose the concept of a surface-attributed graph to include the face attributes, specifically tailored for polyhedron contexts. This representation extends the traditional graph-based approach used in polygon representation by incorporating face-hyperedges in the graph, which encapsulate the geometric properties of a polyhedron's faces.

**Definition 4.1** (Surface-attributed graph). A surface-attributed graph $G = (V, E, F, a)$ is a directed graph, where $V$ is the set of nodes, $E$ is the set of edges, and surface $F$ is the set of face-hyperedges. Each node $v_i = (x_i, y_i, z_i) \in V$ corresponds to a vertex of the polyhedron and is defined by its

coordinates, $x_i, y_i, z_i$ are the values of coordinates. Each directed edge $e_{i,j} = (v_i, v_j) \in E$ represents an edge of a face in the polyhedron. Each face-hyperedge $f = (e_{1,2}, e_{2,3}, ..., e_{N_{b,i},1}) \in F$ is a an ordered set of edges that forms a closed shape, associated with a set of face attributes $a(f)$. It is important to note that, unlike traditional hyperedges in a graph, which simply connect multiple nodes, face-hyperedges contain the connectivity order information of edges, which captures the hierarchical topology of a polyhedron.

**Constructing SAG from a polyhedron**. Based on the discussion so far, we summarize the steps for constructing the SAG from a given polyhedron $q$ as follows: We treat the vertex set in the original polyhedron as the node set $V$ of SAG. Then for a face $p_i$ in the polyhedron $q$, consider each pair of consecutive vertices $(v_j, v_{j+1})$ as the endpoints of an edge $e_{j,j+1} = (v_j, v_{j+1})$. Doing so for $j = 1, ..., N_{b,i} - 1$, and adding an edge between the last and first vertices of $p_i$ to ensure a closed boundary, we will have all the edges of this face. Hence a face-hyperedge is formed as: $f = (e_{1,2}, e_{2,3}, ..., e_{N_{b,i},1})$. Doing this for all faces, we have $F$. we build a mapping $a$ from each $f$ to its attributes. Union all the edges generated from all the faces to form $E$.

By incorporating face-hyperedges, SAG provides a comprehensive representation of a polyhedron that captures all its vertex-level, edge-level and face-level properties. It is important to highlight that SAG inherently captures the adjacency information between faces. From this structural representation, two key observations can be made: 1) If two faces $f_i$ and $f_j$ are adjacent in the polyhedron, their adjacent edges share the same nodes but in opposite directions, such that $\exists e_{o,r} \in f_i, e_{r,o} \in f_j$. 2) Each edge in a face must have a corresponding opposite edge, which belongs to another face. $\forall e_{o,r} \in f_i, \exists e_{r,o} \in f_j, i \neq j$.

**Lemma 4.2.** *Let $q = \{p_i\}_{i=1}^{N_f}$ be a polyhedron and $G = (V, E, F, a)$ be the SAG derived from $q$. The transformation from $q$ to $G$ is invertible.*

*Proof.* The detailed proof is in Appendix B. □

## 4.2 LOCAL RIGID REPRESENTATION OF SAG

The geometric information in a SAG is encapsulated by the relative positions of nodes and the specific shape of each face, which are defined by the node coordinates and connection topology. So attaining a representation for the whole SAG requires the above local information, namely local rigid representation, to be preserved (as elaborated in this subsection) and then be aggregated with minimal information loss (as detailed in Section 4.3).

To achieve local rigid representation, relying solely on node coordinates is insufficient, as this does not preserve essential symmetries such as translation and rotation invariance. Moreover, calculating the distances to all other nodes is computationally expensive and overlook crucial topological features such as edges and faces. Tackling this issue motivates us to seek to encode the relative position of a node through its local rigid, including its neighbor nodes, edges, and faces. Hence, we propose a novel five-tuple geometric representation that maintains the relative positioning of nodes within the graph while also respecting the integrity of its edges and faces. We transform the absolute coordinates of a node into a vector, and once all other nodes are fixed, the position of the target node is determined by its representation.

**Definition 4.3** (Two-hop Path). For a node $v_i$ in a SAG, a two-hop path $\pi_{i,j,k}$ is an ordered sequence of three nodes $(v_i, v_j, v_k)$ where $v_j$ is adjacent to both $v_i$ and $v_k$. We denote the set of all two-hop paths converging to node $v_i$ as $\Pi_2^i$.

**Definition 4.4** (Local Rigid Representation of SAG). The SAG can be expressed as a collection of Local Rigid Representation tuples $s(\pi_{i,j,k})$ as shown in Figure 2:

$$G = \{s(\pi_{i,j,k}) | \pi_{i,j,k} \in \Pi_2^i, v_i \in V\}, \tag{1}$$
$$s(\pi_{i,j,k}) = (d_{i,j}, d_{j,k}, \theta_{i,j,k}, \phi_{i,j,k}, \psi_{i,j,k})$$

where $d_{i,j}$ is the Euclidean distance between node $v_i$ and $v_j$, $d_{j,k}$ is the distance between node $v_j$ and $v_k$, $\theta_{i,j,k} \in [-\pi, \pi)$ is the angle at $v_j$ formed by the three nodes. $\phi_{i,j,k} \in [-\pi, \pi)$ is the dihedral angle between the two faces containing edge $e_{i,j}$ and $e_{j,k}$ respectively, $\psi_{i,j,k}$ denotes the indices of the face-hyperedge containing $e_{i,j}$ and $e_{j,k}$.

Importantly, the representation is invariant under rotation and translation transformations, ensuring that the structural integrity of the graph is maintained regardless of its orientation or position. We further affirm that this representation encapsulates all information of the graph. So by incorporating the local rigid representation of each node, the network would be able to capture the global information of the whole graph as the layer number grows. In essence, utilizing the local rigid representation of the SAG, as detailed in Equation 1, enables us to reconstruct a graph that is equivalent to the original.

**Theorem 4.5.** *Given the local rigid representation of a surface-attributed graph $G$, as articulated in Equation 1, one can reconstruct a graph that is equivalent to $G$.*

*Proof.* The foundational concept of Theorem 4.5 is that faces within a polyhedron are interconnected via shared edges. We first prove that starting from a random node, one can recover the shape of a face it associated with. Then one can iteratively combine the faces to reconstruct an equivalent SAG. The detailed proof is in Appendix C. □

### 4.3 POLYHEDRONGNN ARCHITECTURE

After obtaining the local rigid representations in the previous section, in this section, the second step of our approach solves the problem of aggregating them to obtain a global representation. Specifically, we propose PolyhedronGNN, which operates on the surface-attributed graph $\mathcal{G} = (V, E, F, a)$ and learns to aggregate information from neighboring nodes and faces with a focus on utilizing different models to learn the different interactions in SAG.

In each layer, we utilize the local rigid representation and face attributes to guide the node embedding updating process. As shown in Figure 2 (c), considering a two-hop path $\pi_{i,j,k}$, the consisting edges can be within the same face or different faces. The flow of information from one face to another is critical in learning the interrelation between faces, while intra-face flow enhances the understanding of shapes of a single face. We divide possible path types into two categories: $\psi(\pi_{i,j,k}) \in \{R_{inner}, R_{cross}\}$. To distinguish between different paths, we propose a heterogeneous function for learning the message based on the path type. Let $\Psi^{(l,\psi(\pi_{i,j,k}))}$ be a multi-layer perceptron (MLP) model for path type $\psi(\pi_{i,j,k})$ at layer $l$, the learned message $m^{(l)}(\pi_{i,j,k})$ from path $\pi_{i,j,k}$ can be formulated as follows:

$$m^{(l)}(\pi_{i,j,k}) = w^{(\psi(\pi_{i,j,k}))} \Psi^{(l,\psi(\pi_{i,j,k}))}(h_i^{(l)}, h_j^{(l)}, h_k^{(l)}, g^{(l)}), \tag{2}$$

where $w^{(\psi(\pi_{i,j,k}))}$ is the weight for path type $\psi(\pi_{i,j,k})$, $g^{(l)} = \varphi^{(l)}(d_{i,j}\|d_{j,k}\|\theta_{i,j,k}\|\phi_{i,j,k}\|a_{j,i}\|a_{k,j})$ is the guiding embedding calculated by an MLP function $\varphi^{(l)}$, where $\|$ denotes the concatenation operation, $a_{j,i}, a_{k,j}$ are the face attributes of the faces containing $e_{j,i}, e_{k,j}$, respectively. We initialize node embeddings to zeroes. For a node $v_i$, let $h_i^{(l+1)}$ represent its updated embedding in the $l$-th layer. The node embedding update is formulated as follows:

$$h_i^{(l+1)} = \sum \{m^{(l)}(\pi_{i,j,k}) | \pi_{i,j,k} \in \Pi_2^i\}, \tag{3}$$

To maximize discriminative power, the embeddings of all nodes are summed to form a graph embedding, and the graph embeddings from all layers are concatenated as the final graph representation $h_G$ for downstream tasks:

$$h_G = \|_{l=1}^{L} \left( \sum_{i=1}^{|V|} h_i^{(l)} \right), \tag{4}$$

where $L$ is the number of GNN layers. PolyhedronGNN utilizes local rigid representation to achieves rotation and translation invariance, while retaining the ability to distinguish different graphs. Assuming the distance between any two nodes is bounded within a range, we demonstrate that our method can aggregate complete graph information with arbitrary precision:

**Theorem 4.6.** *Suppose $\eta : \mathcal{S} \to \mathbb{R}$ be a continuous set function with respect to the Hausdorff distance $d_H(\cdot, \cdot)$. Let $S \in \mathcal{S}$ be the set of all two-hop paths of a surface-attributed graph $G$, $S = \{s(\pi_{i,j,k}) | v_i \in V\}$, $\forall \epsilon > 0, \exists K \in \mathbb{Z}$, such that for any $S \in \mathcal{S}$,*

$$|\eta(S) - \zeta(\eta'(S))| < \epsilon, \tag{5}$$

*where $\zeta$ is a continuous function, and $\eta'(S) \in \mathbb{R}^K$ is the output of our proposed method.*

*Proof.* The detailed proof is in Appendix D. Similar to PointNet, in the worst case, our method divides the space into small granules. With a sufficiently large output dimension, our method maps each input into a unique granule. □

# 5 EXPERIMENTS

We evaluate the effectiveness of our approach through two fundamental tasks—classification and retrieval—across four datasets. We first introduce the datasets and comparison methods then provide the main results and analysis. For detailed information on implementation specifics, please see Appendix E.

## 5.1 DATASET

We employ the following datasets for both classification and retrieval tasks, detailed as follows: **MNIST-C**: This dataset contains 13,742 samples of digit polyhedra. We transform 2D polygon shapes from the MNIST-P dataset (Jiang et al., 2019) into 3D by stretching them along the z-axis. Each digit is color-coded (purple for the bottom face, red for the front face, green for side faces excluding the bottom, and blue for the back face) and randomly rotated in 3D space to highlight directional identification. **Building**: Comprising 5,000 polyhedra, this dataset extends 2D polygons from the OpenStreetMap (OSM) building dataset (Yan et al., 2021) into 3D polyhedra. Each building is categorized into one of ten standard alphabetic shapes based on its shape. Unlike MNIST-C, these samples are not subjected to random rotations due to the original lack of alignment. **ShapeNet-P**: Derived from the ShapeNetCore dataset (Chang et al., 2015), this dataset features 2,122 polyhedra across 15 object categories. We employ a mesh merge algorithm to combine coplanar meshes with identical properties into polyhedral objects. Files that still retain numerous mesh faces after merging are dropped. Random rotations are applied. **ModelNet-P**: This dataset, based on ModelNet40 (Wu et al., 2015a), contains 1,303 polyhedra spanning 14 object categories. The processing is the same as ShapeNet-P, including applying random rotations.

## 5.2 COMPARISON METHOD

**ResNet1D** (Mai et al., 2023): This model adapts the 1D variant of the Residual Network (ResNet) architecture, incorporating circular padding to effectively encode the exterior vertices of polygons. **VeerCNN** van't Veer et al. (2019): A Convolutional Neural Network (CNN) designed for 1D inputs, VeerCNN employs zero padding and concludes with global average pooling. **NUFT-DDSL** (Jiang et al., 2019): A spatial domain polygon encoder that uses NUFT features and the DDSL model. **NUFT-IFFT** (Mai et al., 2023): A spatial domain polygon encoder that utilizes NUFT features and the inverse Fast Fourier transformation (IFFT). **PolygonGNN** (Yu et al., 2024): A graph-based polygon encoder that models 2D multipolygon as visibility graph.

## 5.3 EFFECTIVENESS ANALYSIS FOR CLASSIFICATION TASK

Table 1 presents the performance comparison between the proposed method and competing models across four datasets. We utilized a range of metrics to assess performance, including Accuracy (Acc), Weighted Precision (Prec), Weighted F1 Score (F1), and Weighted ROC AUC Score (AUC). The highest scores for each dataset are denoted in boldface. PolyhedronNet achieved the highest scores in accuracy, precision, F1, and AUC across all datasets, enhancing the Precision score by 72% over the average of other methods in the MNIST-C dataset. Notably on the Building dataset, PolyhedronNet achieved an AUC of 1.000. For ShapeNet-P and ModelNet-P, where the challenge lies in handling a diverse range of complex 3D shapes and fine-grained object differences, PolyhedronNet still achieved solid results, with an AUC of 0.936 on ShapeNet-P and 0.824 on ModelNet-P. Although the performance on these datasets was slightly lower compared to MNIST-C and Building, the results still demonstrate its robustness in recognizing complex polyhedra. Overall, PolyhedronNet's performance across these diverse datasets underscores its versatility and strength in handling complex polyhedra, making it an effective solution for the challenging polyhedron classification task.

## 5.4 EFFECTIVENESS ANALYSIS FOR RETRIEVAL TASK

We repurpose the model trained on the classification task to execute the retrieval task by removing the downstream classifier and assessing the similarity among learned representations in the test set. For each test sample, we pre-determine the count of items within the same class and retrieve an equivalent number of samples. We then compute the average values for the following metrics: Precision (Prec),

Table 1: The performance of the proposed model and the comparison methods on the classification task. The best results are in bold.

| Dataset | Metric | NUFT-DDSL | ResNet1D | NUFT-IFFT | VeerCNN | PolygonGNN | PolyhedronNet |
|---------|--------|-----------|----------|-----------|---------|------------|---------------|
| MNIST-C | Acc ↑ | 0.148 | 0.152 | 0.239 | 0.127 | 0.435 | **0.858** |
|         | Prec↑ | 0.092 | 0.139 | 0.220 | 0.104 | 0.446 | **0.861** |
|         | F1↑ | 0.102 | 0.083 | 0.202 | 0.084 | 0.427 | **0.856** |
|         | AUC↑ | 0.474 | 0.610 | 0.619 | 0.576 | 0.801 | **0.985** |
| Building | Acc↑ | 0.921 | 0.919 | 0.941 | 0.874 | 0.973 | **0.980** |
|         | Prec↑ | 0.921 | 0.921 | 0.942 | 0.876 | 0.974 | **0.980** |
|         | F1↑ | 0.921 | 0.920 | 0.941 | 0.874 | 0.973 | **0.980** |
|         | AUC↑ | 0.994 | 0.993 | 0.997 | 0.987 | 0.999 | **1.000** |
| ShapeNet-P | Acc↑ | 0.097 | 0.179 | 0.097 | 0.163 | 0.573 | **0.627** |
|         | Prec↑ | 0.103 | 0.142 | 0.082 | 0.158 | 0.589 | **0.640** |
|         | F1↑ | 0.092 | 0.147 | 0.083 | 0.148 | 0.570 | **0.625** |
|         | AUC↑ | 0.555 | 0.625 | 0.564 | 0.639 | 0.916 | **0.936** |
| ModelNet-P | Acc↑ | 0.153 | 0.321 | 0.164 | 0.206 | 0.430 | **0.435** |
|         | Prec↑ | 0.118 | 0.381 | 0.148 | 0.221 | 0.370 | **0.377** |
|         | F1↑ | 0.114 | 0.302 | 0.138 | 0.197 | 0.385 | **0.393** |
|         | AUC↑ | 0.575 | 0.784 | 0.629 | 0.726 | 0.821 | **0.824** |

Table 2: The performance of the proposed model and the comparison methods on the retrieval task. The best results are in bold.

| Dataset | Metric | NUFT-DDSL | ResNet1D | NUFT-IFFT | VeerCNN | PolygonGNN | PolyhedronNet |
|---------|--------|-----------|----------|-----------|---------|------------|---------------|
| MNIST-C | Prec ↑ | 0.428 | 0.448 | 0.367 | 0.307 | 0.386 | **0.713** |
|         | Recall↑ | 0.430 | 0.450 | 0.368 | 0.308 | 0.388 | **0.715** |
|         | F1↑ | 0.429 | 0.449 | 0.368 | 0.307 | 0.387 | **0.714** |
|         | MAP↑ | 0.660 | 0.696 | 0.559 | 0.477 | 0.586 | **0.842** |
|         | NDCG↑ | 0.897 | 0.910 | 0.857 | 0.809 | 0.859 | **0.945** |
| Building | Prec ↑ | 0.279 | 0.264 | 0.276 | 0.147 | 0.788 | **0.838** |
|         | Recall↑ | 0.282 | 0.266 | 0.279 | 0.148 | 0.796 | **0.847** |
|         | F1↑ | 0.280 | 0.265 | 0.277 | 0.148 | 0.792 | **0.843** |
|         | MAP↑ | 0.564 | 0.481 | 0.550 | 0.327 | 0.890 | **0.923** |
|         | NDCG↑ | 0.809 | 0.771 | 0.803 | 0.645 | 0.953 | **0.966** |
| ShapeNet-P | Prec ↑ | 0.098 | 0.156 | 0.088 | 0.135 | 0.317 | **0.322** |
|         | Recall↑ | 0.101 | 0.161 | 0.091 | 0.139 | 0.327 | **0.332** |
|         | F1↑ | 0.100 | 0.158 | 0.089 | 0.137 | 0.322 | **0.327** |
|         | MAP↑ | 0.201 | 0.299 | 0.196 | 0.291 | 0.476 | **0.486** |
|         | NDCG↑ | 0.415 | 0.525 | 0.405 | 0.513 | 0.670 | **0.674** |
| ModelNet-P | Prec ↑ | 0.113 | 0.196 | 0.118 | 0.155 | 0.233 | **0.240** |
|         | Recall↑ | 0.119 | 0.206 | 0.123 | 0.163 | 0.245 | **0.252** |
|         | F1↑ | 0.116 | 0.201 | 0.120 | 0.159 | 0.239 | **0.246** |
|         | MAP↑ | 0.286 | 0.378 | 0.266 | 0.343 | 0.415 | **0.421** |
|         | NDCG↑ | 0.450 | 0.557 | 0.440 | 0.517 | 0.575 | **0.576** |

Recall, F1 Score (F1), Mean Average Precision (MAP), and Normalized Discounted Cumulative Gain (NDCG).

Table 2 presents the performance comparison between the proposed method and competing models across four datasets. The highest scores for each dataset are denoted in boldface. On the Building dataset, PolyhedronNet exhibited the most significant improvement, with Recall increasing by an average of 60% and F1 showing a substantial boost compared to other methods. It also achieved the highest in other scores, reflecting its ability to retrieve and rank relevant architectural structures accurately. In the MNIST-C dataset, PolyhedronNet outperformed other models, with Precision improving by 32% over the average of other methods, showcasing its effectiveness in retrieving polyhedral representations of handwritten digits. For the ShapeNet-P dataset, which involves distinguishing a wide variety of 3D shapes, PolyhedronNet delivered strong performance, achieving the top NDCG of 0.674, indicating its ability to retrieve and rank relevant shapes effectively. Similarly, in the ModelNet-P dataset, PolyhedronNet excelled, achieving the best NDCG of 0.576, further proving its capacity to handle fine-grained differences in 3D object retrieval. These results demonstrate the versatility and robustness of the representations learned by PolyhedronNet in handling polyhedra.

## 5.5 ABLATION STUDY

We conducted an ablation study to assess the importance of face attributes quantitatively. This involved masking the face attributes with zeroes and comparing the performance to that of the original PolyhedronNet on two specific tasks using the MNIST-C and ShapeNet-P datasets. It is important to note that the Building and ModelNet-P datasets do not possess face attributes, making such comparisons inapplicable. The outcomes of this study are detailed in Table 3 and Table 4. Results demonstrate a noticeable decrease in both classification and retrieval tasks, which indicates the importance of face attributes.

Table 3: Ablation results in classification task

|  | MNIST-C | | ShapeNet-P | |
| --- | --- | --- | --- | --- |
| Metric | w/ face | w/o face | w/ face | w/o face |
| Acc ↑ | 0.858 | 0.360 | 0.627 | 0.578 |
| Prec ↑ | 0.861 | 0.401 | 0.640 | 0.595 |
| F1 ↑ | 0.856 | 0.343 | 0.625 | 0.568 |
| AUC ↑ | 0.985 | 0.742 | 0.936 | 0.909 |

Table 4: Ablation results in retrieval task

|  | MNIST-C | | ShapeNet-P | |
| --- | --- | --- | --- | --- |
| Metric | w/ face | w/o face | w/ face | w/o face |
| Prec ↑ | 0.713 | 0.348 | 0.322 | 0.318 |
| Recall ↑ | 0.715 | 0.349 | 0.332 | 0.327 |
| F1 ↑ | 0.714 | 0.348 | 0.327 | 0.322 |
| MAP ↑ | 0.842 | 0.534 | 0.486 | 0.482 |
| NDCG ↑ | 0.945 | 0.837 | 0.674 | 0.674 |

## 5.6 HYPERPARAMETER SENSITIVITY

We delve into the sensitivity analysis of two critical hyperparameters within our proposed framework: the hidden dimension and the number of GNN layers, utilizing the MNIST-C dataset for evaluation. The impact of the hidden dimension on model performance is illustrated in Figure 3 (a). Generally, the model exhibits low sensitivity to the hidden dimension size once it surpasses a certain threshold (in this case, 256 for the MNIST-C dataset). Nonetheless, dimensions that are too small may constrict the model's expressive capacity, resulting in suboptimal performance. These findings are consistent with the principles outlined in our Theorem 4.6. Regarding the number of GNN layers, Figure 3 (b) shows that an optimal

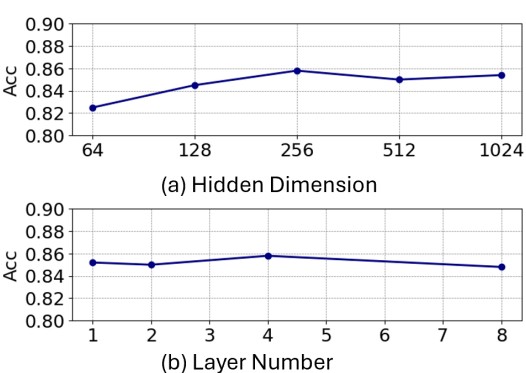

Figure 3: Hyperparameter sensitivity

performance is achieved with approximately 4 GNN layers. The flat curve indicates low sensitivity to the number of layers. This may be attributed to the concatenation of embeddings from all layers.

## 5.7 CASE STUDY

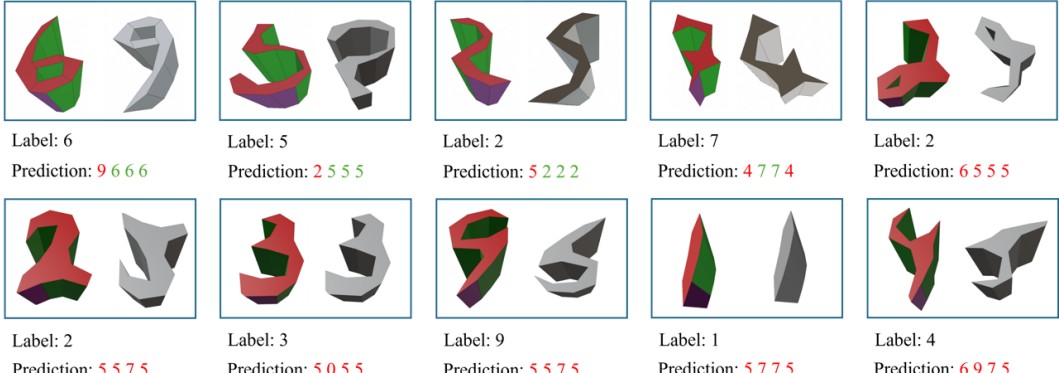

Figure 4: Test cases from the MNIST-C dataset correctly predicted by PolyhedronNet, displaying face-attributed and blank versions side by side. The blank models are rotated to show the possible ambiguity. Predictions from comparison methods are also presented below each image for comparison.

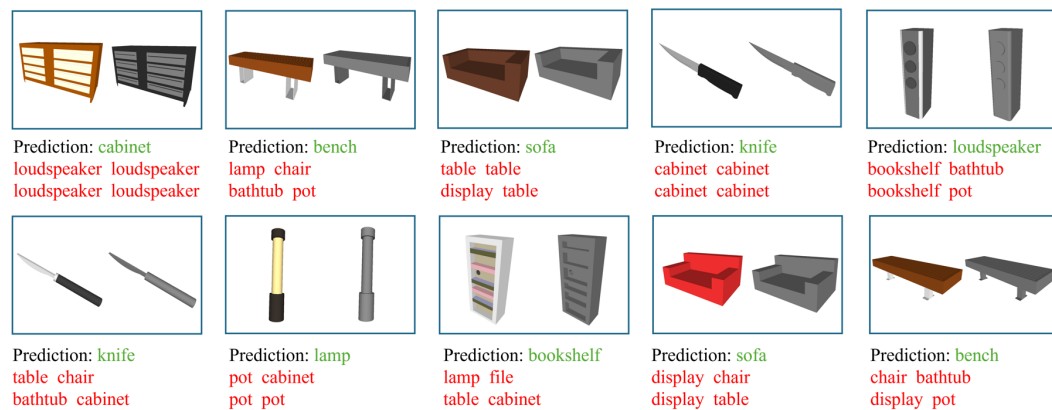

Figure 5: Test cases from the ShapeNet-P dataset correctly predicted by PolyhedronNet, displaying face-attributed and blank versions side by side. Predictions from comparison methods are also presented below each image for comparison.

We conducted an in-depth analysis of PolyhedronNet's performance by selecting and visualizing several representative cases from the test sets of the MNIST-C and ShapeNet-P datasets. The selected cases demonstrate instances where PolyhedronNet's predictions align with the actual labels, and we also present comparative results from other methods for reference. The visualizations from the MNIST-C dataset are depicted in Figure 4. We observed that numerous prediction errors by comparison methods were likely due to the ambiguity caused by rotating polyhedron digits, which can make digits such as '6'/'9' and '5'/'2' appear inverted or flipped. The face attributes within our PolyhedronNet model play a crucial role in indicating the direction of a digit, thereby effectively avoiding such errors. Furthermore, PolyhedronNet demonstrated its ability to accurately handle complex cases where the orientation of digits could lead to misidentification. For instance, it correctly identified an irregularly shaped '7' that resembles a '4' (fourth sample in the first row), a '9' that appeared similar to a '5' (third sample in the second row), and a '4' that resembles a '5' (last sample in the second row). These successes can be partially attributed to the directional guidance provided by face attributes and also to the strong capabilities of our model.

Further visualizations from the ShapeNet-P dataset are shown in Figure 5. In the first case, all comparison methods mistakenly classified the "cabinet" as a "loudspeaker," a common error due to their similar cubic shapes and appearances. However, PolyhedronNet distinguishes the cabinet effectively by recognizing the different colors on its surface, which indicate the presence of drawers, thus negating the possibility of it being a loudspeaker. By adeptly leveraging both the face attributes and the geometric properties of objects, PolyhedronNet enhances prediction accuracy. The ability to discern different parts of objects through attributes like color is particularly effective in complex cases involving multi-part objects such as loudspeakers, knives, lamps, and benches, facilitating accurate feature assembly.

## 6 CONCLUSION

This work advances polyhedra representation learning by introducing a novel framework named PolyhedronNet. Central to this framework is the surface-attributed graph, a unified data structure for modeling polyhedra, coupled with the development of a local rigid representation and a custom-designed graph neural network, PolyhedronGNN. By directly modeling a polyhedron with SAG, we open the door for a variety of applications that require processing 3D polyhedral objects. The effectiveness of PolyhedronNet has been rigorously validated through extensive experiments on four datasets in classification and retrieval tasks.

### ACKNOWLEDGMENTS

This work was partially supported by the NSF Grant No. 2007716, No. 2007976, No. 1942594, No. 1907805, Cisco Faculty Research Award, and NIH Grant No. R01AG089806, No. R01CA297856.

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

## A LIST OF SAMBOLS

The main mathematical symbols used throughout the paper are summarized in Table5, organized with their formal descriptions.

| Symbol | Description |
|---|---|
| $q$ | A polyhedron (3D solid formed by flat polygon faces) |
| $p_i$ | A polygon face in a polyhedron |
| $v_{i,j}$ | The $j$-th vertex of the $i$-th face, with 3D coordinates |
| $N_{b,i}$ | Number of vertices in the $i$-th face |
| $N_f$ | Number of faces in a polyhedron |
| $q_v$ | Vector representation of a polyhedron |
| $G$ | Surface-attributed graph (SAG) comprising $(V, E, F, a)$ |
| $V$ | Set of nodes in the SAG |
| $E$ | Set of edges in the SAG |
| $F$ | Set of face-hyperedges in the SAG |
| $f$ | Face-hyperedge, an ordered set of edges forming a closed shape |
| $a$ | Face attributes mapping function |
| $e_{i,j}$ | Directed edge from vertex $v_i$ to $v_j$ |
| $\pi_{i,j,k}$ | Two-hop path $v_i \leftarrow v_j \leftarrow v_k$ |
| $\Pi_2^i$ | Set of all two-hop paths converging to node $v_i$ |
| $d_{i,j}$ | Euclidean distance between nodes $v_i$ and $v_j$ |
| $\theta_{i,j,k}$ | Angle at $v_j$ formed by vectors $\overrightarrow{v_j v_i}$ and $\overrightarrow{v_j v_k}$ |
| $\phi_{i,j,k}$ | Dihedral angle between faces containing edges $e_{i,j}$ and $e_{j,k}$ |
| $\psi_{i,j,k}$ | Indices of face-hyperedge containing edges $e_{i,j}$ and $e_{j,k}$ |
| $\Psi^{(l,\psi(\pi_{i,j,k}))}$ | Multi-layer perceptron model for path type $\psi(\pi_{i,j,k})$ at layer $l$ |
| $\psi(\pi_{i,j,k})$ | Path type indicator ($R_{inner}$ or $R_{cross}$) |
| $w^{(\psi(\pi_{i,j,k}))}$ | Weight for path type $\psi(\pi_{i,j,k})$ |
| $a_{j,i}$ | Face attributes of the face containing edge $e_{j,i}$ |
| $g^{(l)}$ | Guiding embedding at layer $l$ |
| $\varphi^{(l)}$ | MLP function for calculating the guiding embedding at layer $l$ |
| $h_i^{(l)}$ | Node embedding of node $v_i$ at layer $l$ |
| $m^{(l)}(\pi_{i,j,k})$ | Learned message from path $\pi_{i,j,k}$ at layer $l$ |
| $h_G$ | Final graph representation |

Table 5: Key symbols and their descriptions

## B PROOF FOR LEMMA 4.2

*Proof.* The nodes in graph $G$ have a one-to-one correspondence with the vertices of the polyhedron $q$. Each polygon face $p_i$ is defined by an ordered set of points. To reconstruct $q$ from $G$, we first group the nodes of the graph into their corresponding faces using the face-hyperedge. Since nodes along the boundaries of faces are arranged in a counterclockwise direction when viewed from outside the polyhedron, we can reconstruct the boundary of each face by initiating traversal from any node and following the edges until the starting node is reached. This allows for straightforward identification of face boundaries through basic geometric computations. The normal vector associated with each face can be computed using the cross product of two edges on the face boundary. Consequently, every face is reconstituted with its correct shape and orientation. Hence, from graph $G$, we can uniquely

reconstruct the original polyhedron $q$, ensuring that no information about the polyhedron's structure is lost. □

## C  PROOF FOR THEOREM 4.5

We first proof that starting from a random node, one can recover the shape of a face it associated with.

**Lemma C.1.** *Given the position of a starting node $v_i$, which is connected to $v_j$, and local rigid representations of SAG, we can determine the face shape whose starting edge is $e_{i,j}$ (i.e., $f = (e_{i,j}...)$) in a 2D plane.*

*Proof.* We establish a local 2D Cartesian coordinate system with node $v_j$ as the origin and the ray $\overrightarrow{v_i v_j}$ as the positive x-axis. The coordinate of node $v_i$ now is $(-d_{i,j}, 0)$. Define the angle $\theta_{i,j,k}$ as the clockwise rotation from the ray $\overrightarrow{v_j v_i}$ to the ray $\overrightarrow{v_j v_k}$. A positive value of $\theta_{i,j,k}$ indicates a clockwise rotation, while a negative value indicates a counterclockwise rotation. Given this setup, the coordinates of node $v_k$ relative to $v_j$ can be calculated using trigonometric relations:

$$\begin{cases} x_k = -d_{j,k}\cos(\theta_{i,j,k}), \\ y_k = d_{j,k}\sin(\theta_{i,j,k}) \end{cases}$$

Therefore, by applying these trigonometric relations, we can uniquely determine the coordinates of $v_k$ in the local coordinate system. Then iteratively we can determine the coordinate of the next node following $v_k$ until we reach the starting node $v_i$ to form a closed shape. Since $\psi_{i,j,k}$ indicates whether two consecutive edges belong to the same face, this helps prevent deviations to different faces. Hence, the shape of the face is determined and the lemma is thereby proven. □

Then we prove that we can combine faces to reconstruct an equavalent SAG.

*Proof.* An equivalent SAG is one that represents the same polyhedron, under any translation or rotation transformations. Without loss of generality, we start from a random node as delineated in Lemma C.1, then the first face shape can be determined. Given that the faces within a polyhedron are interconnected through shared edges, we can iteratively apply this process to determine the shapes of all faces in the graph. Since $\phi_{i,j,k}$ records the angles between two associated faces, we can connect two faces by first using $\psi_{i,j,k}$ to identify the adjacent faces, then querying their shared edges, and setting the faces to form an angle equal to $\phi_{i,j,k}$ at the shared edges. By repeating this process iteratively, the position of all faces are determined. It's noteworthy that different initializations, which might lead to varying orientations or positions of the graph due to rotation or translation transformations, still correspond to the same multipolygon. Consequently, despite these transformations, the reconstructed graph retains its equivalence to the original heterogeneous visibility graph. Hence, the theorem is proven. □

## D  PROOF FOR THEOREM 4.6

*Proof.* Since $\eta : \mathcal{S} \to \mathbb{R}$ is a continuous set function with respect to Hausdorff distance, $\forall \epsilon_1 > 0, \exists \delta_1 > 0$ such that for any $S, S' \in \mathcal{S}$ with $d_H(S, S') < \delta_1$, we have $|\eta(S) - \eta(S')| < \epsilon_1$. Assume, without loss of generality, that $S$ is a one-dimensional finite set contained within an interval $[a, b]$. Denote this interval as $\Xi = [a, b]$, we can divide $\Xi$ into $K = \lceil \frac{b-a}{\delta} \rceil + 1$ equal subintervals $[a + (k-1)\Delta, a + k\Delta], k = 1, 2, \ldots, K$, where $\Delta = \frac{b-a}{K}$. Define a function $r : \mathbb{R} \to \mathbb{R}$ as $r(x) = a + \lfloor \frac{x-a}{\Delta} \rfloor \Delta$, which maps each $x \in S$ to the lower bound of its respective subinterval. Let $S' = \{r(x) : x \in S\}$. By this construction, $d_H(S, S') \leq \frac{b-a}{K} < \delta_1$, hence $|\eta(S) - \eta(S')| < \epsilon_1$.

Next, define $\sigma_k : \mathbb{R} \to [0, +\infty)$ as the Hausdorff distance from any point $x$ to the complement of the $k$-th subinterval in $\Xi$. Specifically, $\sigma_k(x) = d_H(x, \Xi \backslash [a + (k-1)\Delta, a + k\Delta])$. Let symmetric function $v_k(S) = \sum_{x \in S} \sigma_k(x)$, indicating whether points of $S$ fall within the $k$-th subinterval.

With these definitions, we construct a mapping function $\tau : [0, +\infty)^K \to \mathcal{S}$ as $\tau(\mathbf{v}) = \{a + (k-1)\Delta : v_k > 0\}$, which maps the vector $\mathbf{v} = [v_1, \ldots, v_K]$ to a set consisting of the lower bounds of the subintervals occupied by $S$, which exactly equals the set $S'$ constructed above, i.e., $\tau(\mathbf{v}(S)) = S'$.

Let $\zeta : \mathbb{R}^K \to \mathbb{R}$ be a continuous function so that $\zeta(\mathbf{v}) = \eta(\tau(\mathbf{v}))$. Denote $\boldsymbol{\sigma} = [\sigma_1, \ldots, \sigma_K]$. Then we have

$$|\eta(S) - \zeta(\sum\{\boldsymbol{\sigma}(x) : x \in S\})|$$
$$=|\eta(S) - \eta(\tau(\sum\{\boldsymbol{\sigma}(x) : x \in S\}))|$$
$$=|\eta(S) - \eta(\tau(\mathbf{v}(S)))|$$
$$=|\eta(S) - \eta(S')| < \epsilon_1$$

The continuous function $\boldsymbol{\sigma}$ can be approximated by a multilayer perceptron, according to the universal approximation theorem. Therefore, We have $|\eta(S) - \zeta(\sum\{m(x) : x \in S\})| < \epsilon$, where $m$ is the MLP function. Considering the method described in Section 4.3, we can set $L = 1$, making our proposed function $\eta'$ a sum of the messages from an MLP function. The sum operator is a special case of our method when $L = 1$ and the message function is the MLP used above. Thus, we arrive at the conclusion that $|\eta(S) - \zeta(\eta'(S))| < \epsilon$. Hence, the theorem is proven. $\qquad\square$

## E    EXPERIMENTAL DETAILS

### E.1    IMPLEMENTATION DETAILS

Each dataset is randomly split into 60%, 20%, and 20% for training, validation, and testing respectively.

We use CrossEntropyLoss as the loss function for all classification tasks. Adam optimizer and ReduceLROnPlateau scheduler are used to optimize the model. The learning rate is set to 0.001 across all tasks and models. The training batch and testing batch are set to 32 for the MNIST-C and Building datasets and 8 for the ShapeNet-P and ModelNet-P datasets. The downstream task model is a four-layer MLP function with batchnorm enabled for the classification task. All models are trained for a maximum of 500 epochs using an early stop scheme.

For the comparison method ResNet1D, VeerCNN, NUFT-DDSL, and NUFT-IFFT, we follow the original settings provided by the authors.

For the message encoding function $\Psi$, we use a four-layer MLP function with batchnorm enabled across all tasks. For the guiding embedding function $\varphi$, we leverage a one-layer MLP function with batchnorm enabled across all tasks. The downstream task classifier is a four-layer MLP function.

The hyperparameters we tuned include hidden dimensions in 64,128,256,512,1024, and the number of GNN layers in 1,2,3,4,8. We found the best hyperparameters for different datasets are: MNIST-C: [256,4]; Building: [512,4]; ShapeNet-P: [256,2]; ModelNet-P: [128,2].

## F    PERFORMANCE OF MORE COMPARISON METHODS AND ABLATED MODELS

We conduct additional experiments to evaluate our method against more comparison methods and ablated models. First, we compare different aggregation strategies, including mean and max aggregators, where results show that max aggregation generally achieves superior performance on MNIST-C and ShapeNet-P datasets, while mean aggregation performs better on Building and ModelNet-P datasets. To validate the effectiveness of our proposed heterogeneous geometric message passing mechanism, we conduct an ablation study (our w/o hetero) where we remove the heterogeneous message-passing modules. The significant performance drop demonstrates the importance of these components in capturing complex geometric relationships. We also compare our method with established graph learning methods, including HGT and HAN. The results show that our approach substantially outperforms these methods across all datasets, indicating the advantage of our design. Furthermore, we benchmark against recent state-of-the-art point cloud methods, including LocoTrans Chen et al. (2024) and RISurConv Zhang et al. (2024). The experimental results demonstrate that our method achieves superior performance on most datasets, particularly showing significant improvements on MNIST-C and Building datasets. This suggests that our approach better captures the inherent geometric structure of 3D shapes compared to point-based methods.

Table 6: The performance of additional ablated models and comparison methods on the classification task.

| Dataset | Metric | our agg_mean | our agg_max | our w/o hetero | HGT | HAN | LocoTrans | RISurConv |
|---|---|---|---|---|---|---|---|---|
| MNIST-C | Acc↑ | 0.858 | 0.885 | 0.754 | 0.113 | 0.221 | 0.344 | 0.567 |
| | Prec↑ | 0.868 | 0.890 | 0.784 | 0.103 | 0.157 | 0.403 | 0.590 |
| | F1↑ | 0.859 | 0.885 | 0.742 | 0.093 | 0.159 | 0.350 | 0.617 |
| | AUC↑ | 0.984 | 0.990 | 0.956 | 0.528 | 0.597 | 0.714 | 0.833 |
| Building | Acc↑ | 0.981 | 0.973 | 0.797 | 0.897 | 0.900 | 0.949 | 0.949 |
| | Prec↑ | 0.981 | 0.973 | 0.821 | 0.899 | 0.902 | 0.950 | 0.929 |
| | F1↑ | 0.981 | 0.973 | 0.768 | 0.896 | 0.899 | 0.949 | 0.937 |
| | AUC↑ | 1.000 | 0.999 | 0.922 | 0.991 | 0.987 | 0.996 | 0.991 |
| ShapeNet-P | Acc↑ | 0.587 | 0.620 | 0.486 | 0.201 | 0.137 | 0.540 | 0.265 |
| | Prec↑ | 0.596 | 0.648 | 0.506 | 0.161 | 0.130 | 0.591 | 0.261 |
| | F1↑ | 0.571 | 0.618 | 0.463 | 0.155 | 0.113 | 0.542 | 0.274 |
| | AUC↑ | 0.919 | 0.921 | 0.900 | 0.651 | 0.622 | 0.886 | 0.730 |
| ModelNet-P | Acc↑ | 0.576 | 0.508 | 0.374 | 0.359 | 0.374 | 0.561 | 0.347 |
| | Prec↑ | 0.581 | 0.508 | 0.327 | 0.360 | 0.371 | 0.546 | 0.343 |
| | F1↑ | 0.564 | 0.500 | 0.333 | 0.300 | 0.333 | 0.539 | 0.332 |
| | AUC↑ | 0.895 | 0.891 | 0.796 | 0.799 | 0.796 | 0.892 | 0.793 |

Table 7: The performance of additional ablated models and comparison methods on the retrieval task.

| Dataset | Metric | our agg_mean | our agg_max | our w/o hetero | HGT | HAN | LocoTrans | RISurConv |
|---|---|---|---|---|---|---|---|---|
| MNIST-C | Prec↑ | 0.690 | 0.717 | 0.579 | 0.284 | 0.389 | 0.417 | 0.533 |
| | Recall↑ | 0.692 | 0.720 | 0.581 | 0.285 | 0.390 | 0.419 | 0.535 |
| | F1↑ | 0.691 | 0.718 | 0.580 | 0.285 | 0.389 | 0.418 | 0.533 |
| | MAP↑ | 0.828 | 0.847 | 0.751 | 0.455 | 0.592 | 0.594 | 0.710 |
| | NDCG↑ | 0.940 | 0.945 | 0.915 | 0.799 | 0.865 | 0.860 | 0.927 |
| Building | Prec↑ | 0.819 | 0.806 | 0.602 | 0.291 | 0.265 | 0.823 | 0.403 |
| | Recall↑ | 0.828 | 0.814 | 0.608 | 0.294 | 0.268 | 0.831 | 0.405 |
| | F1↑ | 0.824 | 0.810 | 0.605 | 0.293 | 0.266 | 0.828 | 0.405 |
| | MAP↑ | 0.910 | 0.908 | 0.756 | 0.549 | 0.509 | 0.912 | 0.665 |
| | NDCG↑ | 0.964 | 0.960 | 0.904 | 0.803 | 0.776 | 0.965 | 0.827 |
| ShapeNet-P | Prec↑ | 0.262 | 0.323 | 0.238 | 0.269 | 0.272 | 0.322 | 0.170 |
| | Recall↑ | 0.271 | 0.333 | 0.246 | 0.278 | 0.281 | 0.332 | 0.182 |
| | F1↑ | 0.267 | 0.328 | 0.242 | 0.273 | 0.277 | 0.326 | 0.173 |
| | MAP↑ | 0.462 | 0.499 | 0.406 | 0.483 | 0.492 | 0.497 | 0.387 |
| | NDCG↑ | 0.664 | 0.690 | 0.614 | 0.688 | 0.693 | 0.686 | 0.602 |
| ModelNet-P | Prec↑ | 0.334 | 0.334 | 0.234 | 0.318 | 0.326 | 0.378 | 0.201 |
| | Recall↑ | 0.351 | 0.351 | 0.246 | 0.334 | 0.342 | 0.396 | 0.199 |
| | F1↑ | 0.342 | 0.342 | 0.240 | 0.326 | 0.334 | 0.386 | 0.204 |
| | MAP↑ | 0.528 | 0.517 | 0.406 | 0.514 | 0.521 | 0.554 | 0.366 |
| | NDCG↑ | 0.680 | 0.670 | 0.568 | 0.673 | 0.678 | 0.683 | 0.539 |

# G  VISUALIZATION OF RETRIEVED OBJECTS FROM SHAPENET-P

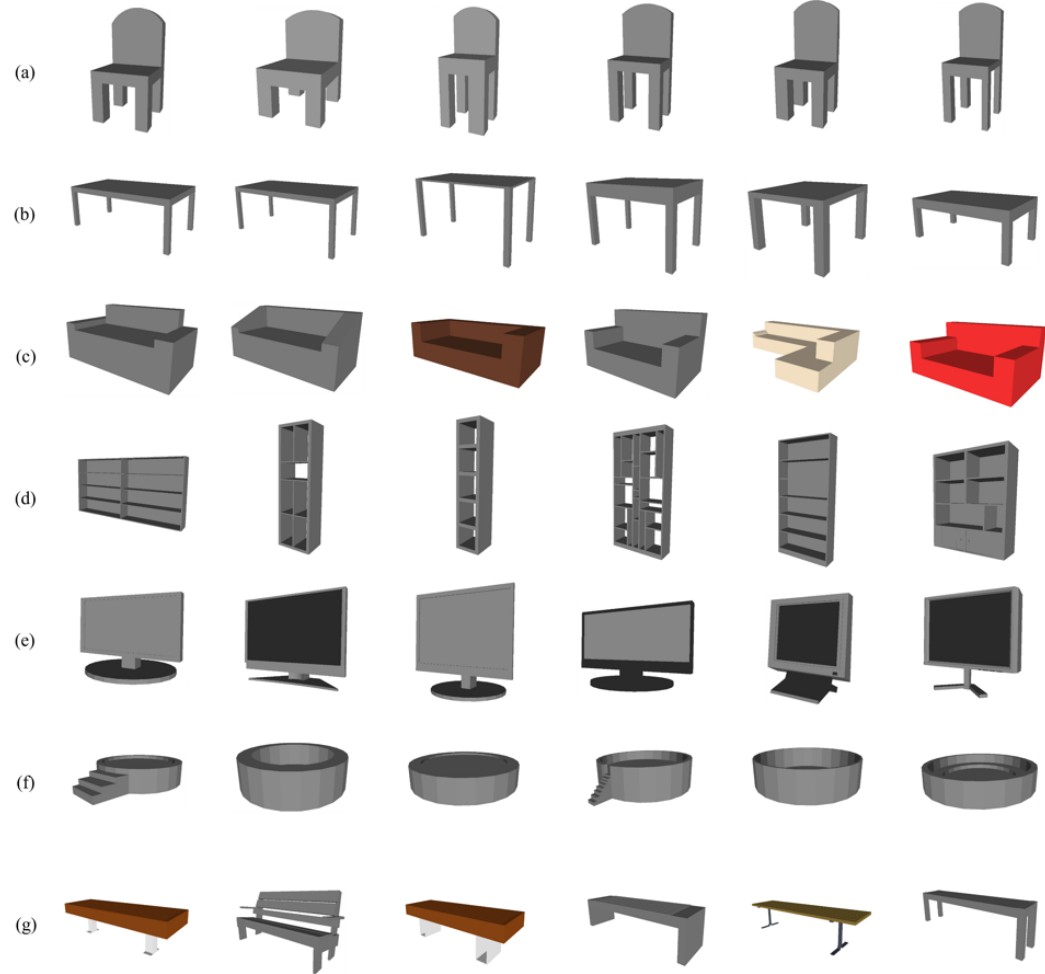

Figure 6: Retrieved samples from ShapeNet-P dataset by PolyhedronNet, the first column shows the query object.

