# OpenReview forum: "PolyhedronNet: Representation Learning for Polyhedra with Surface-attributed Graph"
_ICLR.cc/2025/Conference — ICLR 2025 Poster_

### Official Review · Reviewer_3fMs · 2024-10-31

**Soundness:** 3
**Presentation:** 3
**Contribution:** 3
**Rating:** 6
**Confidence:** 3

**Summary:**

In this paper, the authors propose a novel method for  polyhedra representation learning. The design is straightforward and has been proven to be effective.

**Strengths:**

This design is very straightforward, easy to understand, and has been proven to be effective in the experimental part.

**Weaknesses:**

I am not an expert in this domain; however, I have an understanding of the author's design and experiments. My inquiries primarily stem from the following two aspects:
a)There is an absence of a comparison experiment with Reference [1] mentioned below. It appears that both are engaged in polyhedron representation learning.
b)What are the advantages of this face-based representation method in comparison to the point cloud-based representation method, particularly in numerical experiments? (Given that this paper explicitly mentions the shortcomings of the point cloud-based method.)

[1]Yu, D., Hu, Y., Li, Y., & Zhao, L. (2024, August). PolygonGNN: Representation Learning for Polygonal Geometries with Heterogeneous Visibility Graph. In Proceedings of the 30th ACM SIGKDD Conference on Knowledge Discovery and Data Mining (pp. 4012-4022).

**Questions:**

Please add experiments for comparison with PolygonGNN.
Please add experiments for comparison with modern point cloud methods, especially the works in recent years rather than just PointNet.

---

### Official Review · Reviewer_8zui · 2024-11-03

**Soundness:** 3
**Presentation:** 3
**Contribution:** 2
**Rating:** 6
**Confidence:** 4

**Summary:**

The paper presents PolyhedronNet, a novel framework for representation learning of polyhedra using surface-attributed graphs (SAG). The authors aim to address the limitations of existing methods that predominantly focus on vertex sequences, failing to capture the intricate surface characteristics of 3D polyhedral objects. The proposed approach involves decomposing the SAG into local rigid representations, which effectively maintain geometric relationships while minimizing information loss. The authors introduce PolyhedronGNN, a graph neural network designed to hierarchically aggregate these local representations, achieving a global representation that is invariant to rotation and translation. Experimental results on multiple datasets demonstrate the effectiveness of PolyhedronNet in classification and retrieval tasks, significantly outperforming other methods

**Strengths:**

1. This paper incorporates face attributes into the model, enhancing the understanding of geometric and semantic information and improving representation learning.
2. The authors introduce two message passing mechanisms (inner-face and cross-face) that facilitate information flow within and between faces, strengthening local information aggregation and sensitivity to geometric relationships.
3. Results across multiple datasets show that PolyhedronNet achieves notable performance gains in classification and retrieval tasks, validating the proposed methods and their applicability in 3D object representation learning.

**Weaknesses:**

1. While the paper presents experiments on four datasets, the diversity of polyhedral object types is somewhat limited. I recommend that the authors include a broader range of datasets to enhance the generalizability of the results and to provide a more comprehensive evaluation of the model's performance.
2. The proposed PolyhedronGNN appears to be computationally intensive, especially when applied to large polyhedral datasets. It would be beneficial for the authors to discuss the model's scalability and potential optimization strategies for handling larger datasets efficiently, as this would greatly enhance its practical applicability.

**Questions:**

1. I recommend adding additional baseline comparisons, such as HGT, HAN, and PolygonGNN, to provide a more comprehensive evaluation of PolyhedronNet's performance. Additionally, incorporating some datasets like DBSR for validation could enhance the robustness of the results.
2. The paper employs a sum operation for aggregating vertex features. It would be beneficial to provide experimental results for alternative aggregation methods, such as average and max, to examine their impact on model performance.
3. Could you conduct ablation experiments to compare the inner-face and intra-face geometric message passing modules?
4. Please provide an analysis of the time complexity of the proposed methods. If the complexity is too high, are there any strategies or optimizations that could be implemented to address this issue?
5. Given that the experimental results demonstrate significant improvements, can you provide the open-source code?

---

### Official Review · Reviewer_tXyF · 2024-11-08

**Soundness:** 2
**Presentation:** 4
**Contribution:** 2
**Rating:** 6
**Confidence:** 2

**Summary:**

The paper proposes a framework named PolyhedronNet for effective representation learning of 3D polyhedron. By introducing the Surface-Attributed Graph, the method designs a Local Rigid Representation based on Graph Neural Networks and further develops PolyhedronGNN to aggregate these local representations, ultimately achieving a robust global polyhedral representation invariant to rotation and translation. Experimental results demonstrate significant performance improvements on four datasets, particularly in classification and retrieval tasks.

**Strengths:**

- PolyhedronNet introduces the novel concept of Surface-Attributed Graph for polyhedral representation learning, addressing the limitations of traditional graph representations that cannot capture the face attributes of polyhedra.
- The experimental results on four datasets are thorough, demonstrating superior performance in both classification and retrieval tasks compared to existing methods.

**Weaknesses:**

- Some definitions are expressed quite confusingly, making it difficult to accurately understand what the authors intend to convey. For instance, the notation ϕ_(i,j,k) in Equation 1 and the corresponding illustration in Figure 2(b), as well as the description of a_(j,i)and a_(k,j)in the calculation of g^((l))  in Equation 2.
- The article emphasizes the importance of modeling face attributes, but it does not clearly indicate how the attribute set a in the surface-attributed graph G=(V,E,F,a) is obtained.
- While the paper compares with several methods, it would be beneficial to include comparisons with more recent state-of-the-art methods in geometric deep learning.
- The ablation study is limited to the effect of face attributes. Further ablation studies on different components of the framework could provide more insights.

**Questions:**

- As pointed out in Weakness 2, could you please specify how you modeled the face attributes?
- In the experiments on polyhedral digit recognition shown in Figures 4, how do face attributes effectively differentiate similar digits, such as "6" and "9"?
- In section 5.7, you mentioned that "The ability to discern different parts of objects through attributes like color is particularly effective in complex cases involving multi-part objects such as loudspeakers, knives, lamps, and benches, facilitating accurate feature assembly." However, from the ablation studies in Tables 3 and 4, it seems that the inclusion of face attributes does not significantly affect the ShapeNet dataset, but it severely affects the MNIST-C dataset. Why is that?

---

### Official Review · Reviewer_GCC3 · 2024-11-10

**Soundness:** 3
**Presentation:** 3
**Contribution:** 3
**Rating:** 6
**Confidence:** 2

**Summary:**

The paper presents  a novel graph representation for polyhedral shapes and a novel message-passing approach over this representation to perform polyhedral representation learning.
the proposed surface-attributed graphs maintains an hyperedge per face connecting the ordered boundary edges of said face, and then characterize said graph with a local rigid representation where the connectivity and metric structure is encapuled by the series of two-hop paths from each vertex.
The flow of information from one face to another is captured by splitting said paths into inter-face flows where the path changes face, and
 intra-face flow, capturing information of a single face.

**Strengths:**

Novel rand effective representation, relatively novel redefinition of the message-passing over the representation.
Very strong classification and retrieval tasks.
The paper is relatively well written, although in several places key concepts (such as the local rigid representation) are implicitly defined inline, while a separate formal definition would have made the read easier.

**Weaknesses:**

The work has its niche and works well within it.I don't see any clear weaknesses except perhaps the impact within and outside the niche.

**Questions:**

no further question

---

### Comment · Area_Chair_kkDz · 2024-11-25
**Please check the authors' responses**

Dear reviewers,

Could you please check the authors' responses, and post your message for discussion or changed scores?

best,

AC

---

### Meta-Review · Area_Chair_kkDz · 2024-12-20

**Metareview:**

This paper aims to learn the representation of 3D polyhedral objects, by proposing the a PolyhedronNet that models the vertices, edges, faces and their geometric interrelationships. The major technical contributions include the local rigid representation, geometric message passing, for learning the global representations while having rotation and translation invariance. The proposed approach was evaluated for the four datasets, and results showed its effectiveness.  This proposed  PolyhedronNet works on the 3D polyhedral shapes, with some novel designs over this representation, which is a valuable contribution. Considering the overall positive comments after the rebuttal, the paper can be accepted. Since the reviewers suggested on more experimental comparisons and analysis, the paper should include these revisions in the final version.

**Additional Comments On Reviewer Discussion:**

Reviewer GCC3 did not point out significant weakness, and Reviewer tXyF concerns on some definitions and insufficient comparisons with sota methods and ablation studies. Reviewer 8zui commented on the diversity of polyhedral object types in datasets, model's scalability to larger datasets, adding more baselines methods, time complexity, etc. Reviewer 3fMs suggested on more comparisons with the point cloud network methods. The rebuttal mostly addressed these above concerns, and all reviewers rated score of 6.

---

### Decision · Program_Chairs · 2025-01-22

Accept (Poster)